# A Boosting Approach to Reinforcement Learning

## Abstract

We study efficient algorithms for reinforcement learning in Markov decision processes, whose complexity is independent of the number of states. This formulation succinctly captures large scale problems, but is also known to be computationally hard in its general form. Previous approaches attempt to circumvent the computational hardness by assuming structure in either transition function or the value function, or by relaxing the solution guarantee to a local optimality condition.

We consider the methodology of boosting, borrowed from supervised learning, for converting weak learners into an effective policy. The notion of weak learning we study is that of sampled-based approximate optimization of linear functions over policies. Under this assumption of weak learnability, we give an efficient algorithm that is capable of improving the accuracy of such weak learning methods iteratively. We prove sample complexity and running time bounds on our method, that are polynomial in the natural parameters of the problem: approximation guarantee, discount factor, distribution mismatch and number of actions. In particular, our bound does not explicitly depend on the number of states.

A technical difficulty in applying previous boosting results, is that the value function over policy space is not convex. We show how to use a non-convex variant of the Frank-Wolfe method, coupled with recent advances in gradient boosting that allow incorporating a weak learner with multiplicative approximation guarantee, to overcome the non-convexity and attain global optimality guarantees.

## 1 Introduction

The field of reinforcement learning, formally modelled as learning in Markov decision processes (MDP), models the mechanism of learning from rewards, as opposed to examples. Although the case of tabular MDPs is well understood, the main difficulty in applying RL to practice is the size of the state space.

Various techniques have been suggested and applied to cope with very large MDPs. The most common of which is function approximation of either the value or the transition function of the underlying MDP, many times using deep neural networks. Training deep neural networks in the supervised learning model is known to be computationally hard. Therefore reinforcement learning with neural function approximation is also computationally hard in general, and for this reason lacks provable guarantees.

This challenge of finding efficient and provable algorithms for MDPs with large state space is the focus of our study. Previous approaches can be categorized in terms of the structural assumptions made on the MDP to circumvent the computational hardness. Some studies focus on structured dynamics, whereas others on structured value function or policy classes w.r.t. to the dynamics.

In this paper we study another methodology to derive provable algorithms for reinforcement learning: ensemble methods for aggregating weak or approximate algorithms into substantially more accurate solutions. Our method can be thought of as extending the methodology of boosting from supervised learning (Schapire & Freund, 2012) to reinforcement learning. Interestingly, however, our resulting aggregation of weak learners is not linear.

In order to circumvent the computational hardness of solving general MDPs with function approximation, we assumes access to a weak learner: an efficient sample-based procedure that is capable

of generating an approximate solution to any linear optimization objective over the space of policies. We describe an algorithm that iteratively calls this procedure on carefully constructed new objectives, and aggregates the solution into a single policy. We prove that after sufficiently many iterations, our resulting policy is provably near-optimal.

## 1.1 CHALLENGES AND TECHNIQUES

Reinforcement learning is quite different from supervised learning and several difficulties have to be circumvented for boosting to work. Amongst the challenges that the reinforcement learning setting presents, consider the following,

(a) The value function is not a convex or concave function of the policy. This is true even in the tabular case, and even more so if we use a parameterized policy class.

(b) The transition matrix is unknown, or prohibitively large to manipulate for large state spaces. This means that even evaluation of a policy cannot be exact, and can only be computed approximately.

(c) It is unrealistic to expect a weak learner that attains near-optimal value for a given linear objective over the policy class. At most one can hope for a multiplicative and/or additive approximation of the overall value.

Our approach overcomes these challenges by applied several new as well as recently developed techniques. To overcome the nonconvexity of the value function, we use a novel variant of the Frank-Wolfe optimization algorithm that simultaneously delivers on two guarantees. First, it finds a first order stationary point with near-optimal rate. Secondly, if the objective happens to admit a certain gradient domination property, an important generalization of convexity, it also guarantees near optimal value. The application of the nonconvex Frank-Wolfe method is justified due to previous recent investigation of the policy gradient algorithm (Agarwal et al., 2019; 2020a), which identified conditions under which the value function is gradient dominated.

The second information-theoretic challenge of the unknown transition function is overcome by careful algorithmic design: our boosting algorithm requires only samples of the transitions and rewards. These are obtained by rollouts on the MDP.

The third challenge is perhaps the most difficult to overcome. Thus far, the use of the Frank-Wolfe method in reinforcement learning did not include a multiplicative approximation, which is critical for our application. Luckily, recent work in the area of online convex optimization (Hazan & Singh, 2021) studies boosting with a multiplicative weak learner. We make critical use of this new technique which includes a non-linear aggregation (using a 2-layer neural network) of the weak learners. This aspect is perhaps of general interest to boosting algorithm design, which is mostly based on linear aggregation.

## 1.2 OUR CONTRIBUTIONS

Our main contribution is a novel efficient boosting algorithm for reinforcement learning. The input to this algorithm is a weak learning method capable of approximately optimizing a linear function over a certain policy class.

The output of the algorithm is a policy which does not belong to the original class considered. It is rather a non-linear aggregation of policies from the original class, according to a two-layer neural network. This is a result of the two-tier structure of our algorithm: an outer loop of non-convex Frank-Wolfe method, and an inner loop of online convex optimization boosting. The final policy comes with provable guarantees against the class of all possible policies.

Our algorithm and guarantees come in four flavors, depending on the mode of accessing the MDP (two options), and the boosting methodology for the inner online convex optimization problem (two options).

It is important to point out that we study the question from an optimization perspective, and hence, assume the availability of an efficient exploration scheme – either via access to a reset distribution that has some overlap with the state distribution of the optimal policy, or constraining the policy

|  | Supervised weak learner | Online weak learner |  |
|---|---|---|---|
| Episodic model | $C_\infty^6(\Pi)/\alpha^4\varepsilon^5$ | $C_\infty^4(\Pi)/\alpha^2\varepsilon^3$ | $C_\infty = \max_{\pi\in\Pi}\left\|\frac{d^{\pi^*}}{d^\pi}\right\|_\infty$ |
| Rollouts w. $\nu$-resets | $\mathcal{D}_\infty^6/\alpha^4\varepsilon^6$ | $\mathcal{D}_\infty^4/\alpha^2\varepsilon^4$ | $D_\infty = \left\|\frac{d^{\pi^*}}{\nu}\right\|_\infty$ |

Table 1: Sample complexity of the proposed algorithms for different $\alpha$-weak learning models (supervised & online) and modes of accessing the MDP (rollouts & rollouts with reset distribution $\nu$), suppressing polynomial factors in $|A|, 1/(1-\gamma)$. See Theorem 11 for details.

class to policies that explore sufficiently. Such considerations also arise when reducing reinforcement learning to a sequence of supervised learning problems, e.g. Conservative Policy Iteration (Kakade & Langford, 2002) assumes the former. One contribution we make here is to quantitatively differentiate between these two modes of exploration in terms of the rates of convergence they enable for the boosting setting.

### 1.3 RELATED WORK

To cope with prohibitively large MDPs, the method of choice to approximate the policy and transition space are deep neural networks, dubbed "deep reinforcement learning". Deep RL gave rise to beyond human performance in games such as Go, protein folding, as well as near-human level autonomous driving. In terms of provable methods for deep RL, there are two main lines of work. The first is a robust analysis of the policy gradient algorithm (Agarwal et al., 2019; 2020a). Importantly, the gradient domination property of the value function established in this work is needed in order to achieve global convergence guarantees of our boosting method.

The other line of work for provable approaches is policy iteration, which uses a restricted policy class, making incremental updates, such as Conservative Policy Iteration (CPI) (Kakade & Langford, 2002; Scherrer & Geist, 2014), and Policy Search by Dynamic Programming (PSDP)(Bagnell et al., 2003).

Our boosting approach for provable deep RL builds on the vast literature of boosting for supervised learning (Schapire & Freund, 2012), and recently online learning (Leistner et al., 2009; Chen et al., 2012; 2014; Beygelzimer et al., 2015; Jung et al., 2017; Jung & Tewari, 2018). One of the crucial techniques important for our application is the extension of boosting to the online convex optimization setting, with bandit information (Brukhim & Hazan, 2021), and critically with a multiplicative weak learner (Hazan & Singh, 2021). This latter technique implies a non-linear aggregation of the weak learners. Non-linear boosting was only recently investigated in the context of classification (Alon et al., 2020), where it was shown to potentially enable significantly more efficient boosting.

Perhaps the closest work to ours is boosting in the context of control of dynamical systems (Agarwal et al., 2020b). However, this work critically requires knowledge of the underlying dynamics (transitions), which we do not, and cannot cope with a multiplicative approximate weak learner.

The Frank-Wolfe algorithm is extensively used in machine learning, see e.g. (Jaggi, 2013), references therein, and recent progress in stochastic Frank-Wolfe methods (Hassani et al., 2017; Mokhtari et al., 2018; Chen et al., 2018; Xie et al., 2019). Recent literature has applied a variant of this algorithm to reinforcement learning in the context of state space exploration (Hazan et al., 2019).

## 2 PRELIMINARIES

**Optimization.** We say that a differentiable function $f : \mathcal{K} \mapsto \mathbb{R}$ over some domain $\mathcal{K}$ is $L$-smooth with respect to some norm $\|\cdot\|_*$ if for every $x, y \in \mathcal{K}$ we have

$$\left|f(y) - f(x) - \nabla f(x)^\top(y-x)\right| \leq \frac{L}{2}\|x-y\|_*^2.$$

For constrained optimization (such as over $\Delta_A$), the projection $\Gamma : \mathbb{R}^{|A|} \to \Delta_A$ of a point $x$ to onto a domain $\Delta_A$ is

$$\Gamma[x] = \arg\min_{y \in \Delta_A} \|x - y\|.$$

An important generalization of convex function we use henceforth is that of gradient domination,

**Definition 1** (Gradient Domination). A function $f : \mathcal{K} \to \mathbb{R}$ is said to be $(\kappa, \tau, \mathcal{K}_1, \mathcal{K}_2)$-locally gradient dominated (around $\mathcal{K}_1$ by $\mathcal{K}_2$) if for all $x \in \mathcal{K}_1$, it holds that

$$\max_{y \in \mathcal{K}} f(y) - f(x) \ \leq \ \kappa \times \max_{y \in \mathcal{K}_2} \left\{ \nabla f(x)^\top (y - x) \right\} + \tau.$$

**Markov decision process.** An infinite-horizon discounted Markov Decision Process (MDP) $M = (S, A, P, r, \gamma, d_0)$ is specified by: a state space $S$, an action space $A$, a transition model $P$ where $P(s'|s, a)$ denotes the probability of immediately transitioning to state $s'$ upon taking action $a$ at state $s$, a reward function $r : S \times A \to [0, 1]$ where $r(s, a)$ is the immediate reward associated with taking action $a$ at state $s$, a discount factor $\gamma \in [0, 1)$; a starting state distribution $d_0$ over $S$. For any infinite-length state-action sequence (hereafter, called a trajectory), we assign the following value

$$V(\tau = (s_0, a_0, s_1, a_1, \dots)) = \sum_{t=0}^{\infty} \gamma^t r(s_t, a_t).$$

The agent interacts with the MDP through the choice of stochastic policy $\pi : S \to \Delta_A$ it executes, where $\Delta_A$ denotes the probability simplex over $A$. The execution of such a policy induces a distribution over trajectories $\tau = (s_0, a_0, \dots)$ as

$$P(\tau|\pi) = d_0(s_0) \prod_{t=0}^{\infty} (P(s_{t+1}|s_t, a_t)\pi(a_t|s_t)). \tag{1}$$

Using this description we can associate a state $V^\pi(s)$ and state-action $Q^\pi(s, a)$ value function with any policy $\pi$. For an arbitrary distrbution $d$ over $S$, define:

$$Q^\pi(s) = \mathbb{E}\left[ \sum_{t=0}^{\infty} \gamma^t r(s_t, a_t) \Big| \pi, s_0 = s, a_0 = a \right],$$

$$V^\pi(s) = \mathbb{E}_{a \sim \pi(\cdot|s)}\left[ Q^\pi(s, a)|\pi, s \right], \qquad V_d^\pi = \mathbb{E}_{s_0 \sim d}\left[ V^\pi(s)|\pi \right].$$

Here the expectation is with respect to the randomness of the trajectory induced by $\pi$ in $M$. When convenient, we shall use $V^\pi$ to denote $V_{d_0}^\pi$, and $V^*$ to denote $\max_\pi V^\pi$.

Similarly, to any policy $\pi$, one may ascribe a (discounted) state-visitation distribution $d^\pi = d_{d_0}^\pi$.

$$d_d^\pi(s) = (1 - \gamma) \sum_{t=0}^{\infty} \gamma^t \sum_{\tau : s_t = s} P(\tau|\pi, s_0 \sim d)$$

**Modes of Accessing the MDP.** We henceforth consider two modes of accessing the MDP, that are standard in the reinforcement learning literature, and provide different results for each.

The first natural access model is called the **episodic rollout setting.** This mode of interaction allows us to execute a policy, stop and restart at any point, and do this multiple times.

Another interaction model we consider is called **rollout with $\nu$-restarts.** This is similar to the episodic setting, but here the agent may draw from the MDP a trajectory seeded with an initial state distribution $\nu \neq d_0$. This interaction model was considered in prior work on policy optimization Kakade & Langford (2002); Agarwal et al. (2019). The motivation for this model is two-fold: first, $\nu$ can be used to incorporate priors (or domain knowledge) about the state coverage of the optimal policy; second, $\nu$ provides a mechanism to incorporate exploration into policy optimization procedures.

## 3 SETTING: POLICY AGGREGATION AND WEAK LEARNING

Our boosting algorithms henceforth call upon weak learners to generate weak policies, and aggregate these policies in a way that guarantees eventual convergence to optimality. In this section we formalize both components.

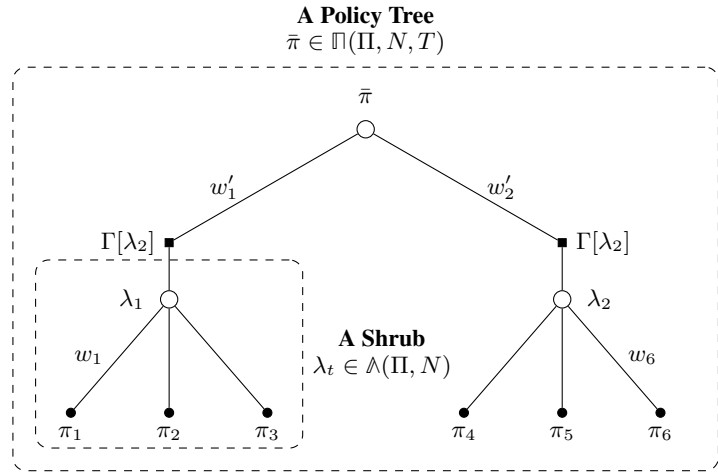

Figure 1: The figure illustrates a Policy Tree hierarchy (see Definition 5), obtained by setting $N = 3$ on the inner loop, and $T = 2$ on the outer loop, to overall get all base policies $\pi_1, ..., \pi_6 \in \Pi_W$ on the lower level. The middle level holds $T = 2$ Policy Shurbs (see Definition 4), where each Shrub $\lambda_t \in \mathbb{A}(\Pi, N)$ is an aggregation of base policies. The top level is an weighted aggregation of the *projected* shrubs $\Gamma[\lambda_t]$, which forms the overall Policy Tree $\bar{\pi} \in \mathbb{\Pi}(\Pi, N, T)$.

### 3.1 POLICY AGGREGATION

For a base class of policies $\Pi_W$, our algorithm incrementally builds a more expressive policy class by aggregating base policies via both linear combinations and non-linear transformations. In effect, the algorithm produces a finite-width depth-2 circuit over some subset of the base policy class. We start with the simpler linear aggregation.

**Definition 2** (Function Aggregation). Given some $N_0 \in \mathbb{Z}_+$, $w \in \mathbb{R}^{N_0}$, $(f_1, \dots f_{N_0}) \in (S \to \mathbb{R}^{|A|})^{\otimes N_0}$, we define $f = \sum_{n=1}^{N_0} w_n f_n$ to be the unique function $f : S \to \mathbb{R}^A$ for which simultaneously for all $s \in S$, it holds

$$f(s) = \sum_{n=1}^{N_0} w_n f(s).$$

Next, the projection operation below may be viewed as a non-linear activation, such as ReLU, in deep learning terms. Note that the projection of any function from $S$ to $\mathbb{R}^{|A|}$ produces a policy, i.e. a mapping from states to distributions over actions.

**Definition 3** (Policy Projection). Given a function $f : S \to \mathbb{R}^{|A|}$, define a projected policy $\pi = \Gamma[f]$ to be a policy such that simultaneously for all $s \in S$, it holds that $\pi(\cdot|s) = \Gamma[f(s)]$.

The next definition defines the class of functions represented by circuits of depth 1 over a base policy class. Note that these function do not necessarily represent policies since they take an affine (vs. convex) combination of policies.

**Definition 4** (Shrub). For an arbitrary base policy class $\Pi \subseteq S \to \Delta_A$, define $\mathbb{A}(\Pi, N)$ to be a set such that $\lambda \in \Lambda(\Pi, N)$ if and only if there exists $N_0 \leq N, w \in \mathbb{R}^{N_0}, (\pi_1, \dots \pi_{N_0}) \in \Pi^{\otimes N_0}$ such that $\lambda = \sum_{n=1}^{N_0} w_n \pi_n$.

The final definition describes the set of possible outputs of the boosting procedure.

**Definition 5** (Policy Tree). For an arbitrary base policy class $\Pi \subseteq S \to \Delta_A$, define $\mathbb{\Pi}(\Pi, N, T)$ to be a policy class such that $\pi \in \mathbb{\Pi}(\Pi, N, T)$ if and only if there exists $T_0 \leq T, w \in \Delta_{T_0}, (\lambda_1, \dots \lambda_{T_0}) \in \mathbb{A}(\Pi, N)^{\otimes T_0}$ such that $\pi = \sum_{t=1}^{T_0} w_t \Gamma[\lambda_t]$.

It is important that the policy that the boosting algorithm outputs can be evaluated efficiently. In the appendix we show it is indeed the case (see Lemma 15).

### 3.2 MODELS OF WEAK LEARNING

We consider two types of weak learners, and give different end results based on the different assumptions: weak supervised and weak online learners. In the discussion below, let $\pi_r$ be a uniformly random policy, i.e. $\forall (s, a) \in S \times A, \pi_r(a|s) = 1/|A|$.

**Supervised Learning.** The natural way to define weak learning is an algorithm whose performance is always slight better than that of random policy, one that chooses an action uniformly at random at any given state. However, in general no learner can outperform a random learner over all label distributions (this is called the "no free lunch" theorem). This motivates the literature on agnostic boosting (Kanade & Kalai, 2009; Brukhim et al., 2020; Hazan & Singh, 2021) that defines a weak learner as one that can approximate the best policy in a given policy class.

**Definition 6** (Weak Supervised Learner). Let $\alpha \in (0, 1)$. Consider a class $\mathcal{L}$ of linear loss functions $\ell : \mathbb{R}^A \to \mathbb{R}$, and $\mathbb{D}$ a family of distributions that are supported over $S \times \mathcal{L}$, policy classes $\Pi_W, \Pi$. A weak supervised learning algorithm, for every $\varepsilon, \delta > 0$, given $m(\varepsilon, \delta)$ samples $D_m$ from any distribution $\mathcal{D} \in \mathbb{D}$ outputs a policy $\mathcal{W}(D_m) \in \Pi_W$ such that with probability $1 - \delta$,

$$\mathbb{E}_{(s,\ell) \sim \mathcal{D}}\big[\ell(\mathcal{W}(D_m))\big] \geq \alpha \max_{\pi^* \in \Pi} \mathbb{E}_{(s,\ell) \sim \mathcal{D}}\big[\ell(\pi^*(s))\big] + (1 - \alpha)\mathbb{E}_{(s,\ell) \sim \mathcal{D}}\big[\ell(\pi_r(s))\big] - \varepsilon.$$

Note that the weak learner outputs a policy in $\Pi_W$ which is approximately competitive against the class $\Pi$. As an additional relaxation, instead of requiring that the weak learning guarantee holds for all distributions, in our setup, it will be sufficient that the weak learning assumption holds over *natural* distributions. We define these below. Hereafter, we refer to $\Pi(\Pi_W, N, T)$ as $\Pi$ for $N, T = O(\text{poly}(|A|, (1 - \gamma)^{-1}, \varepsilon^{-1}, \alpha^{-1}, \log \delta^{-1}))$ specified later.

**Assumption 1** (Weak Supervised Learning). *The booster has access to a weak supervised learning oracle (Definition 6) over the policy class $\Pi$, for some $\alpha \in (0, 1)$. Furthermore, the weak learning condition holds only for a class of* natural *distributions $\mathbb{D} - \mathcal{D} \in \mathbb{D}$ if and only if there exists some $\pi \in \Pi$ such that*

$$\mathcal{D}_S(s) = \int_\ell \mathcal{D}(s, \ell) d\mu(\ell) = d^\pi(s).$$

In particular, while a *natural* distribution may have arbitrary distribution over labels, its marginal distribution over states must be realizable as the state distribution of some policy in $\Pi$ over the MDP $M$. Therefore, the complexity of weak learning adapts to the complexity of the MDP itself. As an extreme example, in stochastic contextual bandits where policies do not affect the distribution of states (say $d_0$), it is sufficient that the weak learning condition holds with respect to all couplings of a single distribution $d_0$.

**Online Learning.** The second model of weak learning we consider requires a stronger assumption, but will give us better sample and oracle complexity bounds henceforth.

**Definition 7** (Weak Online Learner). Let $\alpha \in (0, 1)$. Consider a class $\mathcal{L}$ of linear loss functions $\ell : \mathbb{R}^A \to \mathbb{R}$. A weak online learning algorithm, for every $M > 0$, incrementally for each timestep computes a policy $\mathcal{W}_m \in \Pi_W$ and then observes the state-loss pair $(s, \ell_t) \in S \times \mathcal{L}$ such that

$$\sum_{m=1}^M \ell_m(\mathcal{W}_m(s_m)) \geq \alpha \max_{\pi^* \in \Pi} \sum_{m=1}^M \ell_m(\pi^*(s_m)) + (1 - \alpha) \sum_{m=1}^M \ell_m(\pi_r(s_m)) - R_\mathcal{W}(M).$$

**Assumption 2** (Weak Online Learning). *The booster has access to a weak online learning oracle (Definition 7) over the policy class $\Pi$, for some $\alpha \in (0, 1)$.*

**Remark 8.** A similar remark about *natural* distributions applies to the online weak learner. In particular, it is sufficient the guarantee in 7 holds for arbitrary sequence of loss functions with high probability over the sampling of the state from $d^\pi$ for some $\pi \in \Pi$. Although stronger than supervised weak learning, this oracle can be interpreted as a relaxation of the online weak learning oracle considered in (Brukhim et al., 2020; Brukhim & Hazan, 2021; Hazan & Singh, 2021). A similar model of hybrid adversarial-stochastic online learning was considered in (Rakhlin et al., 2011; Lazaric & Munos, 2009; Beygelzimer et al., 2011). In particular, it is known (Lazaric & Munos, 2009) that unlike online learning, the capacity of a hypothesis class for this model is governed by its VC dimension (vs. Littlestone dimension).

## 4 ALGORITHM & MAIN RESULTS

In this section we describe our RL boosting algorithm. Here we focus on the case where a supervised weak learning is provided. The online weak learners variant of our result is detailed in the appendix. We next define several definitions and algorithmic subroutines required for our method.

**The Extension Operator.** The extension operator (Hazan & Singh, 2021) operate overs functions and modifies their value outside and near the boundary of the convex set $\Delta_A$ to aid the boosting algorithm.

$$F_{G,\beta}[f](x) = \min_{y \in \Delta_A} \left\{ f(y) + G \min_{z \in \Delta_A} \|y - z\| + \frac{1}{2\beta} \|x - y\|^2 \right\}$$

To state the results, we need the following definitions. The first generalizes the policy completeness notion from (Scherrer & Geist, 2014). It may be seen as the policy-equivalent analogue of inherent bellman error (Munos & Szepesvári, 2008). Intuitively, it measures the degree to which a policy in $\Pi$ can best approximate the bellman operator in an average sense with respect to the state distribution induced by a policy from $\Pi$.

**Definition 9** (Policy Completeness). For any initial state distribution $\mu$, define

$$\mathcal{E}_{\mu}(\Pi, \Pi) = \max_{\pi \in \Pi} \min_{\pi^* \in \Pi} \mathbb{E}_{s \sim d_{\mu}^{\pi}} \left[ \max_{a \in A} Q^{\pi}(s, a) - Q^{\pi}(s, \cdot)^{\top} \pi^*(\cdot|s) \right].$$

The following notion of the distribution mismatch coefficient is often useful to characterize the exploration problem faced by policy optimization algorithms.

**Definition 10** (Distribution Mismatch). Let $\pi^* = \arg\max_{\pi} V^{\pi}$, and $\nu$ a fixed initial state distribution (see section 2). Define the following distribution mismatch coefficients:[1]

$$C_{\infty}(\Pi) = \max_{\pi \in \Pi} \left\| \frac{d^{\pi^*}}{d^{\pi}} \right\|_{\infty}, \qquad D_{\infty} = \left\| \frac{d^{\pi^*}}{\nu} \right\|_{\infty}.$$

### 4.1 RL BOOSTING VIA WEAK SUPERVISED LEARNING

We give the main RL boosting algorithm, assuming supervised weak learners. We use a simple sub-routine for choosing a step size, provided in the appendix.

---

**Algorithm 1** RL Boosting via Weak Supervised Learning

---

1: Input parameters $T, N, M, P, \mu$. Initialize a policy $\pi_0 \in \Pi_W$ arbitrarily.
2: **for** $t = 1$ **to** $T$ **do**
3:   Set $\rho_{t,0}$ to be an arbitrary policy in $\Pi_W$.
4:   **for** $n = 1$ **to** $N$ **do**
5:     Execute $\pi_{t-1}$ for $M$ episodes with initial state distribution $\mu$ via Algorithm 2, to get

$$D_{t,n} = \{(s_i, \widehat{Q_i})_{i=1}^m\}.$$

6:     Modify $D_{t,n}$ to produce a new dataset $D'_{t,n} = \{(s_i, f_i)\}_{i=1}^m$, such that for all $i \in [m]$:

$$f_i = -\nabla F_{G,\beta}[-\widehat{Q}_i](\rho_{t,n}(\cdot|s_i))$$

.
7:     Let $\mathcal{A}_{t,n}$ be the policy chosen by the weak learning oracle when given data set $D'_{t,n}$.
8:     Update $\rho_{t,n} = (1 - \eta_{2,n})\rho_{t,n-1} + \frac{\eta_{2,n}}{\alpha}\mathcal{A}_{t,n} - \eta_{2,n}\left(\frac{1}{\alpha} - 1\right)\pi_r$.
9:   **end for**
10:   Declare $\pi'_t = \Gamma[\rho_{t,N}]$.
11:   Choose $\eta_{1,t} = \min\{1, \frac{2C_{\infty}}{t}\}$ if $\mu = d_0$ else $\eta_{1,t} = \texttt{StepChooser}(\pi_{t-1}, \pi'_t, \mu, P)$.
12:   Update $\pi_t = (1 - \eta_{1,t})\pi_{t-1} + \eta_{1,t}\pi'_t$.
13: **end for**
14: Output $\bar{\pi} = \pi_T$ if $\mu = d_0$ else output $\pi_{t-1}$ with the smallest $\eta_t$.

---

[1] For brevity, We use the shorthand $C_{\infty}$ where clear from context.

**Theorem 11.** *Algorithm 1 samples $T(MN + P)$ episodes of length $\tilde{O}(\frac{1}{1-\gamma})$ with probability $1 - \delta$. In the episodic model, for $T = O\left(\frac{C_\infty^2}{(1-\gamma)^3 \varepsilon}\right)$, $N = \left(\frac{16|A|C_\infty}{(1-\gamma)^2 \alpha \epsilon}\right)^2$, $M = m\left(\frac{(1-\gamma)^2 \alpha \varepsilon}{C_\infty |A|}, \frac{\delta}{NT}\right), \mu = d_0$, $P = 0$, with probability $1 - \delta$,*

$$V^* - V^\pi \leq C_\infty \frac{\mathcal{E}(\mathbb{T}, \Pi)}{1 - \gamma} + \varepsilon.$$

*In the $\nu$-reset model, for $T = \frac{8D_\infty^2}{(1-\gamma)^6 \varepsilon^2}$, $N = \left(\frac{16|A|D_\infty}{(1-\gamma)^3 \alpha \epsilon}\right)^2$, $P = \tilde{O}(\frac{200|A|^2 D_\infty^2}{(1-\gamma)^6 \varepsilon^2})$, $M = m\left(\frac{(1-\gamma)^3 \alpha \varepsilon}{8|A|D_\infty}, \frac{\delta}{2NT}\right), \mu = \nu$, with probability $1 - \delta$,*

$$V^* - V^\pi \leq D_\infty \frac{\mathcal{E}_\nu(\mathbb{T}, \Pi)}{(1 - \gamma)^2} + \varepsilon.$$

### 4.2 TRAJECTORY SAMPLER

In Algorithm 2 we describe an episodic sampling procedure, that is used in our sample-based RL boosting algorithms described above. For a fixed initial state distribution $\mu$, and any given policy $\pi$, we apply the following sampling procedure: start at an initial state $s_0 \sim \mu$, and continue to act thereafter in the MDP according to any policy $\pi$, until termination. With this process, it is straightforward to both sample from the state visitation distribution $s \sim d^\pi$, and to obtain unbiased samples of $Q^\pi(s, \cdot)$; see Algorithm 2 for the detailed process.

---

**Algorithm 2** Trajectory Sampler: $s \sim d^\pi$, unbiased estimate of $Q_s^\pi$

---

1: Sample state $s_0 \sim \mu$, and action $a' \sim \mathcal{U}(A)$ uniformly.
2: Sample $s \sim d^\pi$ as follows: at every timestep $h$, with probability $\gamma$, act according to $\pi$; else, accept $s_h$ as the sample and proceed to Step 3.
3: Take action $a'$ at state $s_h$, then continue to execute $\pi$, and use a termination probability of $1 - \gamma$. Upon termination, set $R(s_h, a')$ as the *undiscounted* sum of rewards from time $h$ onwards.
4: Define the vector $\widehat{Q_{s_h}^\pi}$, such that for all $a \in A$, $\widehat{Q_{s_h}^\pi}(a) = |A| \cdot R(s_h, a') \cdot \mathbb{I}_{a=a'}$.
5: **return** $(s_h, \widehat{Q_{s_h}^\pi})$.

---

## 5 ANALYSIS – PROOF SKETCH

We sketch the high-level ideas of the proof of our main result, stated in Theorem 11, and refer the reader to the appendix for the formal proof. Throughout the analysis, we use the notation $\nabla_\pi V^\pi$ to denote the gradient of the value function with respect to the $|S| \times |A|$-sized representation of the policy $\pi$, namely the functional gradient of $V^\pi$.

We establish an equivalence between the outlined algorithm and an abstraction of the Frank-Wolfe algorithm (Algorithm D) from optimization theory. This variant of the Frank-Wolfe (FW) algorithm operates over non-convex and gradient dominated functions to obtain the following novel convergence guarantees. We establish the necessary gradient domination results from the policy completeness results.

**Theorem 12.** *Let $f : \mathcal{K} \to \mathbb{R}$ be $L$-smooth in some norm $\| \cdot \|_*$, $H$-bounded, and the diameter of $\mathcal{K}$ in $\| \cdot \|_*$ be $D$. Then, for a $(\epsilon_0, \mathcal{K}_2)$-linear optimization oracle, the output $\bar{x}$ of Algorithm D satisfies*

$$\max_{u \in \mathcal{K}_2} \nabla f(\bar{x})^\top (u - \bar{x}) \leq \sqrt{\frac{2HLD^2}{T}} + 3\epsilon + \epsilon_0.$$

*Furthermore, if $f$ is $(\kappa, \tau, \mathcal{K}_1, \mathcal{K}_2)$-locally gradient-dominated and $x_0, \dots x_T \in \mathcal{K}_1$, then it holds*

$$\max_{x^* \in \mathcal{K}} f(x^*) - f(\bar{x}) \leq \frac{2\kappa^2 \max\{LD^2, H\}}{T} + \tau + \kappa \epsilon_0.$$

The Frank-Wolfe algorithm utilizes an inner gradient optimization oracle as a subroutine. To implement this oracle using approximate optimizers, we utilize yet another variant of the FW method as "internal-boosting" for the weak learners (by employing an adapted analysis of Theorem 13).

## 5.1 INTERNAL-BOOSTING WEAK LEARNERS

We utilize a variant of the Frank-Wolfe method as a form "internal-boosting" for the weak learners, by employing an adapted analysis of previous work that is stated below.

Note that $\widehat{Q}^\pi(s, \cdot)$ produced by Algorithm 2 satisfies $\|\widehat{Q}^\pi(s, \cdot)\| = \frac{|A|}{1-\gamma}$. We can now borrow the following result on boosting for statistical learning from (Hazan & Singh, 2021), specializing the decision set to be $\Delta_A$. Let $\mathcal{D}_t$ be the distribution induced by the trajectory sampler in round $t$.

**Theorem 13** ((Hazan & Singh, 2021)). *Let $\beta = \sqrt{\frac{1}{\alpha N}}$, and $\eta_{2,n} = \min\{\frac{2}{n}, 1\}$. Then, for any $t$, $\pi'_t$ produced by Algorithm 1 satisfies with probability $1 - \delta$ that*

$$\max_{\pi \in \Pi} \mathbb{E}_{(s,Q) \sim \mathcal{D}_t} \left[ Q^\top \pi(s) \right] - \mathbb{E}_{(s,Q) \sim \mathcal{D}_t} \left[ Q^\top \pi'_t(s) \right] \le \frac{2|A|}{(1-\gamma)\alpha} \left( \frac{2}{\sqrt{N}} + \varepsilon \right)$$

## 5.2 FROM WEAK LEARNING TO LINEAR OPTIMIZATION

In the following Lemma, we give an important observation which allows us to re-state the guarantee in the previous subsection in terms of linear optimization over functional gradients.

**Lemma 14.** *Applying Algorithm 2 for any given policy $\pi$, yields an unbiased estimate of the gradient, such that for any $\pi'$,*

$$(\nabla_\pi V^\pi_\mu)^\top \pi' = \frac{1}{1-\gamma} \mathbb{E}_{(s, \widehat{Q}^\pi(s, \cdot)) \sim \mathcal{D}} \left[ \widehat{Q}^\pi(s, \cdot)^\top \pi'(\cdot|s) \right], \tag{2}$$

*where $\pi'(\cdot|s) \in \Delta_A$, and $\mathcal{D}$ is the distribution induced on the outputs of Algorithm 2, for a given policy $\pi$ and initial state distribution $\mu$.*

*Proof.* Recall $\nabla_\pi V^\pi$ denotes the gradient with respect to the $|S| \times |A|$-sized representation of the policy $\pi$ – the functional gradient. Then, using the policy gradient theorem (Williams, 1992; Sutton et al., 2000), it is given by,

$$\frac{\partial V^\pi_\mu}{\partial \pi(a|s)} = \frac{1}{1-\gamma} d^\pi_\mu(s) Q^\pi(s, a). \tag{3}$$

The following sources of randomness are at play in the sampling algorithm (Algorithm 2): the distribution $d^\pi$ (which encompasses the discount-factor-based random termination, the transition probability, and the stochasticity of $\pi$), and the uniform sampling over $A$. For a fixed $s, \pi$, denote by $\mathcal{Q}^\pi_s$ as the distribution over $\widehat{Q}^\pi(s, \cdot) \in \mathbb{R}^A$, induced by all the aforementioned randomness sources. To conclude the claim, observe that by construction

$$\mathbb{E}_{\mathcal{Q}^\pi(s, \cdot)}[\widehat{Q}^\pi(s, \cdot)|\pi, s] = Q^\pi(s, \cdot). \tag{4}$$

$\square$

## 6 CONCLUSIONS

Building on recent advances in boosting for online convex optimization and bandits, we have described a boosting algorithm for reinforcement learning over large state spaces with provable guarantees. We see this as a first attempt at using a tried-and-tested methodology from supervised learning in RL, and many challenges remain.

First and foremost, our notion of weak learner optimizes a linear function over policy space. A more natural weak learner would be an RL agent with multiplicative optimality guarantee, and it would be interesting to extend our methodology to this notion of weak learnability.

Another important aspect that is not discussed in our paper is that of state-space exploration. Potentially boosting can be combined with state-space exploration techniques to give stronger guarantees independent of distribution mismatch $C_\infty, D_\infty$ factors.

Finally, a feature of our method is that it produces nonlinear aggregations of weak learners as per a two layer neural network. Are simpler aggregations with provable guarantees possible?

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

## A    APPENDIX

It is important that the policy that the boosting algorithm outputs can be evaluated efficiently. Towards that end, we give the following claim.

**Claim 15.** *For any $\pi \in \mathbb{\Pi}(\Pi, N, T)$, $\pi(\cdot|s)$ for any $s \in S$ can be evaluated using $TN$ base policy evaluations and $O(T \times (NA + A \log A))$ arithmetic and logical operations.*

*Proof.* Since $\pi \in \mathbb{\Pi}(\Pi, N, T)$, it is composed of $TN$ base policies. Producing each aggregated function takes $NA$ additions and multiplications; there are $T$ of these. Each projection takes time equivalent to sorting $|A|$ numbers, due to a water-filling algorithm (Duchi et al., 2008); these are also $T$ in number. The final linear transformation takes an additional $TA$ operations. $\square$

## B    STEP-SIZE SUBROUTINE

Below we give an algorithm for choosing step sizes used in both of the RL boosting methods (for online, and supervised, weak learners).

---

**Algorithm 3** StepChooser$(\pi_{t-1}, \pi'_t, \mu, P)$

1: Execute $\pi_{t-1}$ for $P$ episodes with initial state distribution $\mu$ via Algorithm 2, to get

$$D = \{(s_i, \widehat{Q_i})_{i=1}^P\}.$$

2: For any policy $\pi$, let $\widehat{G^\pi} = \frac{1}{P} \sum_{p=1}^P \widehat{Q_i}^\top \pi(\cdot|s_i)$.
3: Return

$$\eta_{1,t} = \texttt{clip}_{[0,1]} \left( \frac{(1-\gamma)^2}{2} \left( \widehat{G^{\pi'_t}} - \widehat{G^{\pi_{t-1}}} \right) \right)$$

---

## C    RL BOOSTING VIA WEAK ONLINE LEARNING

---

**Algorithm 4** RL Boosting via Weak Online Learning

1: Initialize a policy $\pi_0 \in \Pi_W$ arbitrarily.
2: **for** $t = 1$ **to** $T$ **do**
3:     Initialize online weak learners $\mathcal{W}_1, \ldots \mathcal{W}^N$.
4:     **for** $m = 1$ **to** $M$ **do**
5:         Execute $\pi_{t-1}$ once with initial state distribution $\mu$ via Algorithm 2, to get $(s_{t,m}, \widehat{Q}_{t,m})$.
6:         Choose $\rho_{t,m,0} \in \Pi_W$ arbitrarily.
7:         **for** $n = 1$ **to** $N$ **do**
8:             Set $\rho_{t,m,n} = (1 - \eta_{2,n})\rho_{t,m,n-1} + \frac{\eta_{2,n}}{\alpha}\mathcal{W}^n - \eta_{2,n}\left(\frac{1}{\alpha} - 1\right)\pi_r$.
9:         **end for**
10:        Pass to each $\mathcal{W}^n$ the following loss linear $f_{t,m,n}$:

$$f_{t,m,n} = -\nabla F_{G,\beta}[-\widehat{Q}_{t,m}](\rho_{t,m,n}(\cdot|s_i))$$

11:    **end for**
12:    Declare $\pi'_t = \frac{1}{M}\sum_{m=1}^M \Gamma[\rho_{t,m,N}]$.
13:    Choose $\eta_{1,t} = \min\{1, \frac{2C_\infty(\mathbb{\Pi})}{t}\}$ if $\mu = d_0$ else set $\eta_{1,t} = $ StepChooser$(\pi_{t-1}, \pi'_t, \mu, P)$.
14:    Update $\pi_t = (1 - \eta_{1,t})\pi_{t-1} + \eta_{1,t}\pi'_t$.
15: **end for**
16: Output $\bar{\pi} = \pi_T$ if $\mu = d_0$ else output $\pi_{t-1}$ with the smallest $\eta_t$.

---

**Theorem 16.** *Algorithm 4 samples $T(M+P)$ episodes of length $\frac{1}{1-\gamma}\log\frac{T(M+P)}{\delta}$ with probability $1-\delta$. In the episodic model, Algorithm 4 guarantees as long as $T = \frac{16C_\infty^2(\mathbb{\Pi})}{(1-\gamma)^3\varepsilon}$, $N = \left(\frac{16|A|C_\infty(\mathbb{\Pi})}{(1-\gamma)^2\alpha\epsilon}\right)^2$,*

$$M = \max\left\{ \frac{1000|A|^2 C_\infty^2(\Pi)}{(1-\gamma)^4 \varepsilon^2 \alpha^2} \log^2 T\delta, \frac{8|A|C_\infty(\Pi)R_{\mathcal{W}}(M)}{(1-\gamma)^2 \alpha\varepsilon} \right\}, \mu = d_0, \text{ we have with probability } 1 - \delta$$

$$V^* - V^\pi \leq C_\infty(\Pi)\frac{\mathcal{E}(\Pi, \Pi)}{1 - \gamma} + \varepsilon$$

*In the $\nu$-reset model, Algorithm 1 guarantees as long as $T = \frac{100 D_\infty^2}{(1-\gamma)^6 \varepsilon^2}$, $N = \left(\frac{20|A|D_\infty}{(1-\gamma)^3 \alpha\epsilon}\right)^2$,
$P = \frac{250 D_\infty^2 |A|^2}{(1-\gamma)^6 \varepsilon^2} \log^2 \frac{T}{\delta}$, $M = \max\left\{ \left(\frac{40|A|D_\infty}{(1-\gamma)^3 \alpha\varepsilon} \log \frac{T}{\delta}\right)^2, \frac{10|A|D_\infty R_{\mathcal{W}}(M)}{(1-\gamma)^3 \alpha\varepsilon} \right\}, \mu = \nu$, we have with
probability $1 - \delta$*

$$V^* - V^\pi \leq D_\infty \frac{\mathcal{E}_\nu(\Pi, \Pi)}{(1 - \gamma)^2} + \varepsilon$$

*If $R_{\mathcal{W}}(M) = \sqrt{M \log |\mathcal{W}|}$ for some measure of weak learning complexity $|\mathcal{W}|$, the algorithm
samples $\tilde{O}\left(\frac{C_\infty^4(\Pi)|A|^2 \log |\mathcal{W}|}{(1-\gamma)^7 \alpha^2 \varepsilon^3}\right)$ episodes in the episodic model, and $\tilde{O}\left(\frac{D_\infty^4 |A|^2 \log |\mathcal{W}|}{(1-\gamma)^{12} \alpha^2 \varepsilon^4}\right)$ in the $\nu$-
reset model.*

## D NON-CONVEX FRANK-WOLFE

In this section, we give an abstract high-level procedural template that the previously introduced RL boosters operate in. This is based on a variant of the Frank-Wolfe optimization technique, adapted to non-convex and gradient dominated function classes (see Definition 1).

The Frank-Wolfe (FW) method assumes oracle access to a black-box linear optimizer, denoted $\mathcal{O}$, and utilizes it by iteratively making oracle calls with modified objectives, in order to solve the harder task of convex optimization. Analogously, boosting algorithms often assume oracle access to a "weak" learner, which are utilized by iteratively making oracle calls with modified objective, in order to obtain a "strong" learner, with boosted performance. In the RL setting, the objective is in fact non-convex, but exhibits gradient domination. By adapting Frank-Wolfe technique to this setting, we will in subsequent section obtain guarantees for the algorithms given in Section 4.

**Setting.** Denote by $\mathcal{O}$ a black-box oracle to an $(\epsilon_0, \mathcal{K}_2)$-approximate linear optimizer over a convex set $\mathcal{K} \subseteq \mathbb{R}^d$ such that for any given $v \in \mathbb{R}^d$, we have

$$v^\top \mathcal{O}(v) \geq \max_{u \in \mathcal{K}_2} v^\top u - \epsilon_0.$$

---

**Algorithm 5** Non-convex Frank-Wolfe

1: Input: $T > 0$, objective $f$, linear optimization oracle $\mathcal{O}$
2: Choose $x_0$ arbitrarily.
3: **for** $t = 1, \ldots, T$ **do**
4:    Call $z_t = \mathcal{O}(\nabla_{t-1})$, where $\nabla_{t-1} = \nabla f(x_{t-1})$.
5:    Choose $\eta_t = \min\{1, \frac{2\kappa}{t}\}$ in the gradient-dominated case, else choose $\eta_t$ so that

$$|LD^2 \eta_t - \nabla_{t-1}^\top(z_t - x_{t-1})| \leq \epsilon.$$

6:    Set $x_t = (1 - \eta_t)x_{t-1} + \eta_t z_t$.
7: **end for**
8: **return** $\bar{x} = x_T$ in the gradient-dominated case, else $x_{t-1}$ with the smallest $\eta_t$.

---

**Theorem 17.** *Let $f : \mathcal{K} \to \mathbb{R}$ be $L$-smooth in some norm $\|\cdot\|_*$, $H$-bounded, and the diameter of $\mathcal{K}$ in $\|\cdot\|_*$ be $D$. Then, for a $(\epsilon_0, \mathcal{K}_2)$-linear optimization oracle, the output $\bar{x}$ of Algorithm D satisfies*

$$\max_{u \in \mathcal{K}_2} \nabla f(\bar{x})^\top(u - \bar{x}) \leq \sqrt{\frac{2HLD^2}{T}} + 3\epsilon + \epsilon_0.$$

*Furthermore, if $f$ is $(\kappa, \tau, \mathcal{K}_1, \mathcal{K}_2)$-locally gradient-dominated and $x_0, \ldots x_T \in \mathcal{K}_1$, then it holds*

$$\max_{x^* \in \mathcal{K}} f(x^*) - f(\bar{x}) \leq \frac{2\kappa^2 \max\{LD^2, H\}}{T} + \tau + \kappa\epsilon_0.$$

# E    ANALYSIS FOR BOOSTING WITH SUPERVISED LEARNING (PROOF OF THEOREM 11)

**Theorem** (Formal version of Theorem 11). *Algorithm 1 samples $T(MN + P)$ episodes of length $\frac{1}{1-\gamma} \log \frac{T(MN+P)}{\delta}$ with probability $1 - \delta$. In the episodic model, Algorithm 1 guarantees as long as $T = \frac{16C_\infty^2(\Pi)}{(1-\gamma)^3\varepsilon}$, $N = \left(\frac{16|A|C_\infty(\Pi)}{(1-\gamma)^2\alpha\epsilon}\right)^2$, $M = m\left(\frac{(1-\gamma)^2\alpha\varepsilon}{8C_\infty(\Pi)|A|}, \frac{\delta}{NT}\right)$, $\mu = d_0$, we have with probability $1 - \delta$*

$$V^* - V^\pi \;\leq\; C_\infty(\Pi)\frac{\mathcal{E}(\Pi, \Pi)}{1 - \gamma} + \varepsilon$$

*In the $\nu$-reset model, Algorithm 1 guarantees as long as $T = \frac{8D_\infty^2}{(1-\gamma)^6\varepsilon^2}$, $N = \left(\frac{16|A|D_\infty}{(1-\gamma)^3\alpha\epsilon}\right)^2$, $P = \frac{200|A|^2D_\infty^2}{(1-\gamma)^6\varepsilon^2} \log \frac{2TN}{\delta}$, $M = m\left(\frac{(1-\gamma)^3\alpha\varepsilon}{8|A|D_\infty}, \frac{\delta}{2NT}\right)$, $\mu = \nu$, we have with probability $1 - \delta$*

$$V^* - V^\pi \;\leq\; D_\infty\frac{\mathcal{E}_\nu(\Pi, \Pi)}{(1 - \gamma)^2} + \varepsilon$$

*If $m(\varepsilon, \delta) = \frac{\log|\mathcal{W}|}{\varepsilon^2} \log \frac{1}{\delta}$ for some measure of weak learning complexity $|\mathcal{W}|$, the algorithm samples $\tilde{O}\left(\frac{C_\infty^6(\Pi)|A|^4 \log|\mathcal{W}|}{(1-\gamma)^{11}\alpha^4\varepsilon^5}\right)$ episodes in the episodic model, and $\tilde{O}\left(\frac{D_\infty^6|A|^4 \log|\mathcal{W}|}{(1-\gamma)^{18}\alpha^4\varepsilon^6}\right)$ in the $\nu$-reset model.*

*Proof of Theorem 11.* The broad scheme here is to utilize an equivalence between Algorithm 1 and Algorithm D on the function $V^\pi$ (or $V_\nu^\pi$ in the $\nu$-reset model), to which Theorem 17 applies.

To this end, firstly, note $V^\pi$ is $\frac{1}{1-\gamma}$-bounded. Define a norm $\|\cdot\|_{\infty,1} : \mathbb{R}^{|S|\times|A|} \to \mathbb{R}$ as $\|x\|_{1,\infty} = \max_{s\in S} \sum_{a\in A} |x_{s,a}|$. Further, observe that for any policy $\pi : S \to \Delta_A$, $\|\pi\|_{\infty,1} = 1$. The following lemma specifies the smoothness of $V^\pi$ in this norm.

**Lemma 18.** $V^\pi$ *is $\frac{2\gamma}{(1-\gamma)^3}$-smooth in the $\|\cdot\|_{\infty,1}$ norm.*

To be able to interpret Algorithm 1 as an instantiation of the algorithmic template Algorithm D presents, we advance two claims: one, the step-size choices of the two algorithms conincide; two, $\pi_t'$ (Line 3-10) serves as an approximate linear optimizers for $\nabla V^{\pi_{t-1}}$. Together, these imply that the iterates produced by the two algorithms conincide. The first of these, which provides a value of $\epsilon$ to use in the statement of Theorem 17, is established below.

**Claim 19.** *Upon every invocation of* `StepChooser`*, the output $\eta_{1,t}$ satisfies with probability $1 - \delta$*

$$\left|\frac{2\eta_{1,t}}{(1-\gamma)^3} - (\nabla V_\mu^{\pi_{t-1}})^\top(\pi_t' - \pi_{t-1})\right| \;\leq\; \frac{16|A|}{(1-\gamma)^2\sqrt{P}} \log \frac{1}{\delta}$$

Next, we move onto the linear optimization equivalence. Indeed, Claim 20 demonstrates that $\pi_t'$ serves a linear optimizer over gradients of the function $V^\pi$; the suboptimality specifies $\epsilon_0$.

**Claim 20.** *Let $\beta = \sqrt{\frac{1}{\alpha N}}$, and $\eta_{2,n} = \min\{\frac{2}{n}, 1\}$. Then, for any $t$, $\pi_t'$ produced by Algorithm 1 satisfies with probability $1 - \delta$*

$$\max_{\pi\in\Pi}(\nabla V_\mu^{\pi_{t-1}})^\top(\pi - \pi_t') \;\leq\; \frac{2|A|}{(1-\gamma)^2\alpha}\left(\frac{2}{\sqrt{N}} + \varepsilon_W\right)$$

Finally, observe that it is by construction that $\pi_t \in \Pi$. Therefore, in terms of the previous section, $\mathcal{K}$ is the class of all policies, $\mathcal{K}_1 = \Pi, \mathcal{K}_2 = \Pi$.

In the episodic model, we wish to invoke the second part of Theorem 17. The next lemma establishes gradient-domination properties of $V^\pi$ to support this.

**Lemma 21.** $V^\pi$ *is $\left(C_\infty(\Pi), \frac{1}{1-\gamma}C_\infty(\Pi)\mathcal{E}(\Pi, \Pi), \Pi, \Pi\right)$-gradient dominated, i.e. for any $\pi \in \Pi$:*

$$V^* - V^\pi \;\leq\; C_\infty(\Pi)\left(\frac{1}{1-\gamma}\mathcal{E}(\Pi, \Pi) + \max_{\pi'\in\Pi}(\nabla V^\pi)^\top(\pi' - \pi)\right)$$

Deriving $\kappa, \tau$ from the above lemma along with $\epsilon_0$ from Claim 20 and $\epsilon$ from Claim 19, as a consequence of the second part of Theorem 17, we have with probability $1 - NT\delta$

$$V^* - V^{\bar{\pi}} \leq C_\infty(\Pi)\frac{\mathcal{E}(\Pi, \Pi)}{1 - \gamma} + \frac{4C_\infty^2(\Pi)}{(1 - \gamma)^3 T} + \frac{4|A|C_\infty(\Pi)}{(1 - \gamma)^2\alpha\sqrt{N}} + \frac{2|A|C_\infty(\Pi)}{(1 - \gamma)^2\alpha}\varepsilon_W.$$

Similarly, in the $\nu$-reset model, the first part of Theorem 17 provides a local-optimality guarantee for $V_\nu^\pi$. Lemma 22 provides a bound on the function-value gap (on $V^\pi$) provided such local-optimality conditions.

**Lemma 22.** *For any $\pi \in \Pi$, we have*

$$V^* - V^\pi \leq \frac{1}{1 - \gamma}D_\infty\left(\frac{1}{1 - \gamma}\mathcal{E}_\nu(\Pi, \Pi) + \max_{\pi' \in \Pi}(\nabla V_\nu^\pi)^\top(\pi' - \pi)\right)$$

Again, using the bound on $\max_{\pi' \in \Pi}(\nabla V_\nu^{\bar{\pi}})^\top(\pi' - \bar{\pi})$ Theorem 17 provides, we have that with probability $1 - 2NT\delta$

$$V^* - V^{\bar{\pi}} \leq \frac{D_\infty\mathcal{E}_\nu(\Pi, \Pi)}{(1 - \gamma)^2} + \frac{2D_\infty}{(1 - \gamma)^3\sqrt{T}} + \frac{2|A|D_\infty}{(1 - \gamma)^3\alpha}\left(\frac{2}{\sqrt{N}} + \varepsilon_W\right) + \frac{48|A|D_\infty}{(1 - \gamma)^3\sqrt{P}}\log\frac{1}{\delta}$$

$\square$

# F  ANALYSIS FOR BOOSTING WITH ONLINE LEARNING (PROOF OF THEOREM 16)

*Proof of Theorem 16.* Similar to the proof of Theorem 11, we establish an equivalence between Algorithm 1 and Algorithm D on the function $V^\pi$ (or $V_\nu^\pi$ in the $\nu$-reset model), to which Theorem 17 applies provided smoothness (see Lemma 18).

Indeed, Claim 23 demonstrates $\pi_t'$ serves a linear optimizer over gradients of the function $V^\pi$, and provides a bound on $\epsilon_0$. Claim 19 ensures that that the step size choices (and hence iterates) of the two algorithms coincide. As before, observe that it is by construction that $\pi_t \in \Pi$.

**Claim 23.** *Let $\beta = \sqrt{\frac{1}{\alpha N}}$, and $\eta_{2,n} = \min\{\frac{2}{n}, 1\}$. Then, for any $t$, $\pi_t'$ produced by Algorithm 4 satisfies with probability $1 - \delta$*

$$\max_{\pi \in \Pi}(\nabla V_\mu^{\pi_{t-1}})^\top(\pi - \pi_t') \leq \frac{2|A|}{(1 - \gamma)^2\alpha}\left(\frac{2}{\sqrt{N}} + \frac{R_\mathcal{W}(M)}{M} + \sqrt{\frac{16\log\delta^{-1}}{M}}\right)$$

In the episodic model, one may combine the second part of Theorem 17, which provides a bound on function-value gap for gradient dominated functions, which Lemma 21 guarantees, to conclude with probability $1 - T\delta$

$$V^* - V^{\bar{\pi}} \leq \frac{C_\infty(\Pi)\mathcal{E}(\Pi, \Pi)}{1 - \gamma} + \frac{4C_\infty^2(\Pi)}{(1 - \gamma)^3 T} + \frac{4|A|C_\infty(\Pi)}{(1 - \gamma)^2\alpha\sqrt{N}} + \frac{2|A|C_\infty(\Pi)}{(1 - \gamma)^2\alpha}\frac{R_\mathcal{W}(M)}{M} + \frac{8|A|C_\infty(\Pi)\log\delta^{-1}}{(1 - \gamma)^2\alpha\sqrt{M}}.$$

Similarly, in the $\nu$-reset model, Lemma 22 provides a bound on the function-value gap provided local-optimality conditions, which the first part of Theorem 17 provides for. Again, with probability $1 - T\delta$

$$V^* - V^{\bar{\pi}} \leq \frac{D_\infty\mathcal{E}_\nu(\Pi, \Pi)}{(1 - \gamma)^2} + \frac{2D_\infty}{(1 - \gamma)^3}\left(\frac{1}{\sqrt{T}} + \frac{|A|}{\alpha}\left(\frac{2}{\sqrt{N}} + \frac{R_\mathcal{W}(M)}{M} + \frac{4\log\delta^{-1}}{\sqrt{M}}\right) + \frac{24|A|}{\sqrt{P}}\log\frac{1}{\delta}\right)$$

$\square$

## G   PROOFS OF SUPPORTING CLAIMS

### G.1   NON-CONVEX FRANK-WOLFE METHOD (THEOREM 17)

*Proof of Theorem 17.* **Non-convex case.** Note that for any timestep $t$, it holds due to smoothness that

$$f(x_t) = f(x_{t-1} + \eta_t(z_t - x_{t-1}))$$

$$\geq f(x_{t-1}) + \eta_t \nabla_{t-1}^\top(z_t - x_{t-1}) - \eta_t^2 \frac{L}{2} D^2$$

$$= f(x_{t-1}) - \frac{1}{2LD^2}\left(LD^2\eta_t - \nabla_{t-1}^\top(z_t - x_{t-1})\right)^2 + \frac{(\nabla_{t-1}^\top(z_t - x_{t-1}))^2}{2LD^2}$$

Using the step-size definition to bound on the middle term, and telescoping this inequality over function-value differences across successive iterates, we have

$$\min_t(\nabla_{t-1}^\top(z_t - x_{t-1}))^2 \leq \frac{1}{T}\sum_{t=1}^{T}(\nabla_{t-1}^\top(z_t - x_{t-1}))^2 \leq \frac{2LD^2 H}{T} + \epsilon^2$$

Let $t' = \arg\min_t \eta_t$ and $t^* = \arg\min_t(\nabla_{t-1}^\top(z_t - x_{t-1}))^2$. Then

$$\nabla_{t'-1}^\top(z_{t'} - x_{t'-1}) \leq LD^2\eta_{t'} + \epsilon \leq LD^2\eta_{t^*} + \epsilon$$

$$\leq \nabla_{t^*-1}^\top(z_{t^*} - x_{t^*-1}) + 2\epsilon \leq \sqrt{\frac{2LD^2 H}{T} + \epsilon^2} + 2\epsilon$$

To conclude the claim for the non-convex part, observe $\sqrt{a+b} \leq \sqrt{a} + \sqrt{b}$ for $a, b > 0$, and that since $z_{t'} = \mathcal{O}(\nabla_{t'-1})$, it follows by oracle definition that

$$\max_{u \in \mathcal{K}_2} \nabla_{t'-1}^\top u \leq \nabla_{t'-1}^\top z_{t'} + \epsilon_0.$$

**Gradient-Dominated Case.** Define $x^* = \arg\max_{x \in \mathcal{K}} f(x)$ and $h_t = f(x^*) - f(x_t)$.

$$h_t \leq h_{t-1} - \eta_t \nabla_{t-1}^\top(z_t - x_{t-1}) + \eta_t^2 \frac{L}{2} D^2 \qquad \text{smoothness}$$

$$\leq h_{t-1} - \eta_t \max_{y \in \mathcal{K}_2} \eta_t \nabla_{t-1}^\top(y - x_{t-1}) + \eta_t^2 \frac{L}{2} D^2 + \eta_t \epsilon_0 \qquad \text{oracle}$$

$$\leq h_{t-1} - \frac{\eta_t}{\kappa}(f(x^*) - f(x_{t-1})) + \eta_t^2 \frac{L}{2} D^2 + \eta_t \left(\epsilon_0 + \frac{\tau}{\kappa}\right) \qquad \text{gradient domination}$$

$$= \left(1 - \frac{\eta_t}{\kappa}\right) h_{t-1} + \eta_t^2 \frac{L}{2} D^2 + \eta_t \left(\epsilon_0 + \frac{\tau}{\kappa}\right)$$

The theorem now follows from the following claim.

**Claim 24.** *Let $C \geq 1$. Let $g_t$ be a $H$-bounded positive sequence such that*

$$g_t \leq \left(1 - \frac{\sigma_t}{C}\right) g_{t-1} + \sigma_t^2 D + \sigma_t E.$$

*Then choosing $\sigma_t = \min\{1, \frac{2C}{t}\}$ implies $g_t \leq \frac{2C^2 \max\{2D, H\}}{t} + CE.$*

$\square$

### G.2   SMOOTHNESS OF VALUE FUNCTION (LEMMA 18)

*Proof of Lemma 18.* Consider any two policies $\pi, \pi'$. Using the Performance Difference Lemma (Lemma 3.2 in (Agarwal et al., 2019), e.g.) and Equation 2, we have

$$|V^{\pi'} - V^{\pi} - \nabla V^{\pi}(\pi' - \pi)|$$

$$= \frac{1}{1-\gamma}\left|\mathbb{E}_{s \sim d^{\pi'}}\left[Q^{\pi}(\cdot|s)^\top(\pi'(\cdot|s) - \pi(\cdot|s)] - \mathbb{E}_{s \sim d^{\pi}}\left[Q^{\pi}(\cdot|s)^\top(\pi'(\cdot|s) - \pi(\cdot|s)]\right|\right.$$

$$\leq \frac{1}{(1-\gamma)^2}\|d^{\pi'} - d^{\pi}\|_1 \|\pi' - \pi\|_{\infty,1}$$

The last inequality uses the fact that $\max_{s,a} Q^\pi(s,a) \leq \frac{1}{1-\gamma}$. It suffices to show $\|d^{\pi'} - d^\pi\|_1 \leq \frac{\gamma}{1-\gamma}\|\pi' - \pi\|_{\infty,1}$. To establish this, consider the Markov operator $P^\pi(s'|s) = \sum_{a\in A} P(s'|s,a)\pi(a|s)$ induced by a policy $\pi$ on MDP $M$. For any distribution $d$ supported on $S$, we have

$$\|(P^{\pi'} - P^\pi)d\|_1 = \sum_{s'} \left| \sum_{s,a} P(s'|s,a)d(s)(\pi'(a|s) - \pi(a|s)) \right|$$

$$\leq \sum_{s'} P(s'|s,a)\|d\|_1\|\pi' - \pi\|_{\infty,1} \leq \|\pi' - \pi\|_{\infty,1}$$

Using sub-additivity of the $l_1$ norm and applying the above observation $t$ times, we have for any $t$

$$\|((P^{\pi'})^t - (P^\pi)^t)d\|_1 \leq t\|\pi' - \pi\|_{\infty,1}.$$

Finally, observe that

$$\|d^{\pi'} - d^\pi\|_1 \leq (1-\gamma)\sum_{t=0}^\infty \gamma^t \|((P^{\pi'})^t - (P^\pi)^t)d_0\|_1$$

$$\leq \|\pi' - \pi\|_{\infty,1}(1-\gamma)\sum_{t=0}^\infty t\gamma^t = \frac{\gamma}{1-\gamma}\|\pi' - \pi\|_{\infty,1}$$

$\square$

### G.3 STEP-SIZE GUARANTEE (CLAIM 19)

*Proof of Claim 19.* Let $\mathcal{D}$ be the distribution induced by Algorithm 2 upon being given $\pi_{t-1}$. Due to Lemma 14, it suffices to demonstrate that for any $\pi \in \{\pi'_t, \pi_{t-1}\}$ the following claim holds with probability $1 - \frac{\delta}{2}$. The claim in turn follows from Hoeffding's inequality, while noting $\widehat{Q^{\pi_{t-1}}}(s,\cdot)$ is $\frac{|A|}{1-\gamma}$-bounded in the $l_\infty$ norm.

$$\left| \widehat{G^\pi} - \mathbb{E}_{(s,\widehat{Q^{\pi_{t-1}}}(s,\cdot))\sim\mathcal{D}}\left[ \widehat{Q^{\pi_{t-1}}}(s,\cdot)^\top \pi(\cdot|s) \right] \right| \leq \frac{8|A|}{(1-\gamma)\sqrt{P}} \log\frac{1}{2\delta}$$

$\square$

### G.4 GRADIENT DOMINATION (LEMMA 21 AND LEMMA 22)

*Proof of Lemma 21.* Invoking Lemma 4.1 from (Agarwal et al., 2019) with $\mu = d_0$, we have

$$V^* - V^\pi \leq \left\| \frac{d^{\pi^*}}{d^\pi} \right\|_\infty \max_{\pi_0}(\nabla V^\pi)^\top(\pi_0 - \pi)$$

$$\leq C_\infty(\Pi)(\max_{\pi_0}(\nabla V^\pi)^\top\pi_0 - \max_{\pi'\in\Pi}(\nabla V^\pi)^\top\pi' + \max_{\pi'\in\Pi}(\nabla V^\pi)^\top(\pi' - \pi))$$

Finally, with the aid of Equation 2, observe that

$$\max_{\pi_0}(\nabla V^\pi)^\top\pi_0 - \max_{\pi'\in\Pi}(\nabla V^\pi)^\top\pi' = \min_{\pi'\in\Pi}\frac{1}{1-\gamma}\mathbb{E}_{s\sim d^\pi}\left[ \max_a Q^\pi(s,a) - Q^\pi(\cdot|s)^\top\pi' \right]$$

$$\leq \frac{1}{1-\gamma}\mathcal{E}(\Pi,\Pi)$$

$\square$

*Proof of Lemma 22.* Invoking Lemma 4.1 from (Agarwal et al., 2019) with $\mu = \nu$, we have

$$V^* - V^\pi \leq \frac{1}{1-\gamma}\left\| \frac{d^{\pi^*}}{\nu} \right\|_\infty \max_{\pi_0}(\nabla V_\nu^\pi)^\top(\pi_0 - \pi)$$

$$\leq \frac{1}{1-\gamma}D_\infty(\max_{\pi_0}(\nabla V_\nu^\pi)^\top\pi_0 - \max_{\pi'\in\Pi}(\nabla V_\nu^\pi)^\top\pi' + \max_{\pi'\in\Pi}(\nabla V_\nu^\pi)^\top(\pi' - \pi))$$

Again, with the aid of Equation 2, observe that

$$\max_{\pi_0}(\nabla V_\nu^\pi)^\top \pi_0 - \max_{\pi' \in \Pi}(\nabla V_\nu^\pi)^\top \pi' = \min_{\pi' \in \Pi} \frac{1}{1-\gamma} \mathbb{E}_{s \sim d_\nu^\pi} \left[ \max_a Q^\pi(s,a) - Q^\pi(\cdot|s)^\top \pi' \right]$$

$$\leq \frac{1}{1-\gamma} \mathcal{E}_\nu(\Pi, \mathbb{\Pi})$$

$\square$

### G.5 Supervised linear optimization guarantees (Claim 20)

*Proof of Claim 20.* The subroutine presented in lines 3-10 (which culminate in $\pi_t'$) is an instantiation of Algorithm 3 from (Hazan & Singh, 2021), specializing the decision set to be $\Delta_A$. To note the equivalence, note that in (Hazan & Singh, 2021) the algorithm is stated assuming that the center-of-mass of the decision set is at the origin (after a coordinate transform); correspondingly, the update rule in Algorithm 1 can be written as

$$(\rho_{t,n} - \pi_r) = (1 - \eta_{2,n})(\rho_{t,n-1} - \pi_r) + \frac{\eta_{2,n}}{\alpha}(\mathcal{A}_{t,n} - \pi_r).$$

For any state $s$, $\pi_r(\cdot|s) = \frac{1}{A}\mathbf{1}_{|A|}$ corresponds to the center-of-masss of $\Delta_A$. Finally, note that maximizing $f^\top x$ over $x \in \mathcal{K}$ is equivalent to minimizing $(-f)^\top x$ over the same domain. Therefore, we can borrow the following result on boosting for statistical learning from (Hazan & Singh, 2021) (Theorem 13). Note that $\widehat{Q^\pi}(s, \cdot)$ produced by Algorithm 2 satisfies $\|\widehat{Q^\pi}(s, \cdot)\| = \frac{|A|}{1-\gamma}$. Let $\mathcal{D}_t$ be the distribution induced by the trajectory sampler in round $t$.

**Theorem 25** ((Hazan & Singh, 2021)). *Let $\beta = \sqrt{\frac{1}{\alpha N}}$, and $\eta_{2,n} = \min\{\frac{2}{n}, 1\}$. Then, for any $t$, $\pi_t'$ produced by Algorithm 1 satisfies with probability $1 - \delta$ that*

$$\max_{\pi \in \Pi} \mathbb{E}_{(s,Q) \sim \mathcal{D}_t} \left[ Q^\top \pi(s) \right] - \mathbb{E}_{(s,Q) \sim \mathcal{D}_t} \left[ Q^\top \pi_t'(s) \right] \leq \frac{2|A|}{(1-\gamma)\alpha} \left( \frac{2}{\sqrt{N}} + \varepsilon \right)$$

Lemma 14 allows us to restate the guarantees in the previous subsection in terms of linear optimization over functional gradients. The conclusion thus follows immediately by combining Lemma 14 and Theorem 25. $\square$

### G.6 Online linear optimization guarantees (Claim 23)

*Proof of Claim 23.* In a similar vein to the proof of Claim 20, here we state the a result on boosting for online convex optimization (OCO) from (Hazan & Singh, 2021) (Theorem 6), the counterpart of Theorem 13 for the online weak learning case.

**Theorem 26** ((Hazan & Singh, 2021)). *Let $\beta = \sqrt{\frac{1}{\alpha N}}$, and $\eta_{2,n} = \min\{\frac{2}{n}, 1\}$. Then, for any $t$, $\Gamma[\rho_{t,m,N}]$ produced by Algorithm 4 satisfies*

$$\max_{\pi \in \Pi} \sum_{m=1}^{M} \left[ \hat{Q}_{t,m}^\top \pi(s_{t,m}) \right] - \sum_{m=1}^{M} \left[ \hat{Q}_{t,m}^\top \Gamma[\rho_{t,m,N}](s_{t,m}) \right] \leq \frac{2|A|}{(1-\gamma)\alpha} \left( \frac{2M}{\sqrt{N}} + R_\mathcal{W}(M) \right)$$

Next we invoke online-to-batch conversions. Note that in Algorithm 4, $(s_{t,m}, \hat{Q}_{t,m})$ for any fixed $t$ is sampled i.i.d. from the same distribution. Therefore, we can apply online-to-batch results, i.e. Theorem 9.5 in (Hazan, 2019), on Theorem 26 to get

$$\max_{\pi \in \Pi} \mathbb{E}_{(s,Q) \sim \mathcal{D}_t} \left[ Q^\top \pi(s) \right] - \mathbb{E}_{(s,Q) \sim \mathcal{D}_t} \left[ Q^\top \pi_t'(s) \right] \leq \frac{2|A|}{(1-\gamma)\alpha} \left( \frac{2}{\sqrt{N}} + \frac{R_\mathcal{W}(M)}{M} + \sqrt{\frac{16 \log \delta^{-1}}{M}} \right)$$

We finally invoke Lemma 14. $\square$

## G.7   REMAINING PROOFS (CLAIM 24)

*Proof of Claim 24.* Let $T^* = \arg\max_t\{t : t \leq 2C\}$. For any $t \leq T^*$, we have $\sigma_t = 1$ and $g_t \leq H \leq \frac{2C^2H}{t}$. For $t \geq T^*$, we proceed by induction. The base case $(t = T^*)$ is true by the previous display. Now, assume $g_{t-1} \leq \frac{2C^2\max\{2D,H\}}{t-1} + CE$ for some $t > T^*$.

$$
\begin{aligned}
g_t &\leq \left(1 - \frac{2}{t}\right)\left(\frac{2C^2\max\{2D, H\}}{t-1} + CE\right) + \frac{4C^2D}{t^2} + \frac{2CE}{t} \\
&\leq CE + 2C^2\max\{2D, H\}\left(\frac{1}{t-1}\left(1 - \frac{2}{t}\right) + \frac{1}{t^2}\right) \\
&= CE + 2C^2\max\{2D, H\}\frac{t^2 - 2t + t - 1}{t^2(t-1)} \\
&\leq CE + 2C^2\max\{2D, H\}\frac{t(t-1)}{t^2(t-1)}
\end{aligned}
$$

$\square$

