# OpenReview forum: "A Boosting Approach to Reinforcement Learning"
_ICLR.cc/2022/Conference — ICLR 2022 Submitted_

### Official Review · Reviewer_8LJa · 2021-11-03

**Correctness:** 4
**Technical Novelty And Significance:** 3
**Empirical Novelty And Significance:** Not applicable
**Recommendation:** 6
**Confidence:** 3

**Main Review:**

Strengths

- The authors propose an interesting and novel approach to solving the RL problem with unknown transition function and large state space. The use of combining weak learners and leveraging boosting ideas from supervised learning seems significant and new. I believe the core idea will be valuable to both RL and optimization communities.
- The proposed algorithm aims to be agnostic, offering a way to bypass common structural assumptions (e.g. linearity) as long as one has access to certain weak learners. Such a setting has not been explored much previously in RL theory to my knowledge.
- The technical results needed to achieve the sample complexity guarantee may be of independent interest.
- The proofs seem reasonable, but I did not carefully check.


Weaknesses

- I think an example of a valid weak learner that satisfies either the SL or online assumptions would be very helpful for understanding the applicability of the results. For instance, are there existing algorithms that could generate such weak learners easily? Perhaps the answer is obvious, but in any case this seems like an important missing piece.


- The paper is very tersely written and not easy to understand. There is little motivation or justification for both definitions and algorithmic/analytical design choices. Many are simply stated. Even some theorems (11, 13) are just stated. I believe a remark or an interpretation or a comparison with prior work would be enormously helpful. There is no description of any steps in Algorithm 1 besides the algorithm box itself. This issue is also compounded by the fact that there are many notational inaccuracies (see minor comments below for a subset), so a reader trying to parse the approach and results cannot reliably just read the math either.


Specific examples of lack of motivation of *definitions*: “The extension operator… operate overs functions and modifies their value…” Why are we interested in such an operator? Are there other operators we could use in its place?
“To state the results, we need the following definitions” Why do we need these definitions? Are they being used to sweep certain problems under the rug (i.e. Def 10 for exploration)?

I believe these parts (and others throughout) would significantly benefit from more supporting text explaining the reasoning.


Minor comments:
- It would be helpful to also include more discussion about the known achievable PAC sample complexities for RL and MDPs in the tabular case in order to highlight the difference with the state-space independent results here. For example, generative models (not the same setting but close): “Model-Based Reinforcement Learning with a Generative Model is Minimax Optimal” (Agarwal et al, 2019)
- Assumption 1 seems strong since it looks like the weak learner is required to be good on any potential distribution induced by the policies. Can you elaborate on this? An example, as suggested earlier, would be helpful.
- R_W is never defined in the main text.
- $\eta_{2, t}$ is undefined in Alg 1 until Thm 13.
- The input parameters of Alg 1 are not described. One has to look at Alg 3 in the appendix to see what P is even used for.
- M and P are overloaded.
- In the definition of M in Theorem 11, there is a little m followed by a tuple. What is this?
- Little m is undefined in Alg 1.
- Thm 11: $\pi$ is undefined. I think this should say $\bar{\pi}$.
- Thm 11 (and elsewhere): From what I can see, the policy class $\Pi$ is never defined except in some places it is said to be an arbitrary base subset of all policies. When used in the theorem, is this meant to be $\Pi_W$? How should we interpret the term involving $\mathcal{E}$ in Thm 11 (depending on the answer to the previous question)?
- There is no Alg D. I think this should be Alg 5.



**Summary Of The Paper:**

This paper proposes a new approach for solving RL problems with sample complexity independent of the number of states. Rather than imposing structural assumptions, the authors consider access to weak learners and propose a way to combine these weak learners effectively to generate a near optimal policy. The sample complexity result is competitive and does not depend on the number of states, under the assumption of access to weak learners.

**Summary Of The Review:**

I believe the results are interesting, decently significant, and relevant to the community. That being said, there are issues with clarity (noted above). I believe fixing these issues is an easy way to significantly strengthen the paper.

---

> ### Author Response · Authors · 2021-11-19
> **Response**
>
> We thank the reviewer for reading the manuscript and for their feedback and suggestions. We will clarify the comments pointed out by the reviewer in the final version.
>
> - “example of a valid weak learner” - The WL assumption is essentially the assumption that there exists an optimization algorithm for linear functions over the policy space. This is easy to verify for simple policy spaces (e.g., convex spaces), but more challanging for complicated classes (e.g. Deep Neural Networks (DNNs)). We stress that although the weak learnability assumptions in this work cannot be easily verified for complex spaces such as DNNs, we are unaware of any other theory which can make general boosting guarantees without even stricter assumptions. Note that similar "weak learnability" assumptions of DNNs cannot be verified even for renowned classical boosting methods for supervised learning. Nevertheless these methods are used by practitioners and have found practical success. Thus, we view our work as one of the first steps towards provable boosting techniques in RL, which is otherwise notoriously difficult to analyze.
>
> - We agree with the reviewer that the paper is notation-heavy and will add a notation list in the appendix of the paper. We will revise the final version of the paper to elaborate on intuition of the different steps of the algorithm to help the readers get a clearer understanding of our approach.

---

### Official Review · Reviewer_Zawz · 2021-11-08

**Correctness:** 3
**Technical Novelty And Significance:** 3
**Empirical Novelty And Significance:** 1
**Recommendation:** 5
**Confidence:** 3

**Main Review:**

FYI: this is an emergency review that I agreed to do earlier today, so I have not examined the appendix in detail.

The paper is extremely dense and hard to read in many places. It suffers substantially from notation overload. It is very hard to parse the results and contributions since they are obscured by very dense notation. This is further made difficult by a few typos and undefined notation in some places. The paper reads like a list of mathematical concepts / definitions that are not really strung together in a coherent (to the reader) manner. I have no doubt that these concepts are all important to the results, but the paper would be vastly improved by a) improving clarity and doing more to help guide the reader's understanding of what is going on and why it is important and b) reducing the notation significantly, even at the loss of some generality or at the expense of moving some material to the appendix.

This paper would also be improved significantly by a numerical example, even on something very simple like a grid world. I was left with no intuition about how this method would perform in practice. In a similar vein, the paper did not discuss when / if one should use the main proposed algorithm. At the start the authors mention working in 'very large MDPs' and 'MDPs with large state space' - but that is not really supported with any discussion or experiment (other than the complexity bound, which is not enough to build intuition).

The 'two-layer neural network' is mentioned a few times, but I don't really see how it is used / discussed. Where in the algorithm is the two layer NN used?

Not enough discussion is given to the basics of boosting.

I like the inclusion of the proof sketch, but it should be even more detailed and should provide much more intuition.

Several claims are made in the paper without citation - eg the sentence "Training deep neural networks in the supervised learning model is known to be computationally hard". This requires citations not just because it is a controversial claim, but because you say it is 'known' but you don't provide the reader with a way to learn about it if they are unaware of this fact. This pattern is repeated in several places, eg: "However, in general no learner can outperform a random learner over all label distributions (this is called the “no free lunch" theorem)." Please give the reader a reference to learn about this theorem.

A big benefit of the proposed algorithm is that the final complexity bound does not depend on the number of states, which is a very interesting result. However, I have no idea how that came about. The paper should explain how the authors managed to get rid of the dependence on S.

One point I am confused about - "assume the availability of an efficient exploration scheme" - where / how is this assumption used?

The paper says: "we assumes access to a weak learner: an efficient sample-based procedure that is capable of generating an approximate solution to any linear optimization objective over the space of policies" but also says that the problem over policies is non-convex (ie, non-linear). Do you mean to say 'linear optimization objective over the space of **state-visitation distributions**'?

Minor points and typos:

* 'w.r.t.' - don't abbreviate
* "we assumes"
* "weak learner that attains"
* Is the initial state distribution mu or d0?
* Page 4 - What is d_d ? Is s_0 \sim d_0 or d ?
* I do not understand the equation in Definition 2 at all
* "always slight better"
* What is d \mu in Assumption 1? What is D(s, l).
* What is R_W(M) in Definition 7?
* "weak learning is provided"
* bellman -> Bellman
* Several terms in Alg 1 are undefined (eg. \eta_2 never defined, \hat Q never defined). The period is misplaced after line 6.
* Theorem 11:  "Algorithm 1 samples T(MN + P) episodes of length O˜(1/1−γ) with probability 1 − δ." I do not understand what 1-\delta is doing there.
* "Note that Qπ (s, ·) produced by Algorithm 2 satisfies ||Qπ (s, ·)|| =|A|\1−γ." What norm is this? Where is that justified?




**Summary Of The Paper:**

The paper proposes boosting (using Frank-Wolfe) weak RL learners to get a strong (in a concrete sense) policy. The main theoretical bound does not depend on the number of states, which indicates that the method should be competitive in very large state-space MDPs.

**Summary Of The Review:**

An interesting result, however I cannot recommend publication due to significant clarity and readability issues.

---

> ### Author Response · Authors · 2021-11-19
> **Response**
>
> We thank the reviewer for reading the manuscript and for their feedback and suggestions. We will clarify the comments pointed out by the reviewer in the final version.
>
> - We agree with the reviewer that the paper is notation-heavy and will add a notation list in the appendix of the paper. We will revise the final version of the paper to elaborate on intuition of the different steps of the algorithm to help the readers get a clearer understanding of our approach.
> - 'two-layer neural network' - our boosting algorithm yields an aggregated learner that can be thought of as a two-layer neural network (the 2 layers correspond to the 2 loops of Alg.1). See Figure 1 for a depiction of this structure, where the leaves correspond to policies \pi obtained via the weak learner.
> - “One point I am confused about..” - this is implicitly assumed in the usage of the bounded terms C_inf, and D_inf. In particular, consider the followining assumptions:
>   - The learner has access to a reset distribution that is “close” to the optimal policy’s state distribution.
>   - Constraining the policy class to policies that explore sufficiently.
>
>   Notice that these two cases correspond to our “Episodic model” and “rollout with \nu-restarts” settings, or modes of accessing the MDP. In the first setting we assume a bounded C_inf,  and the second assumes bounded D_inf.  Utilizing such assumptions  is required for achieving meaningful results in this context, which is otherwise notoriously difficult to analyze. This follows previous works which also depend on similar assumptions on C_infty, e.g. Agarwal et al. 2019.

---

> > ### Comment · Reviewer_Zawz · 2021-11-30
> > **Response**
> >
> > Thank you to the authors for taking the time to respond to the review, and thank you for clarifying some things I had confusion about. In light of their updates I have improved my score from 3 to 5. I am still very concerned about clarity in the paper overall and I think the paper would be made much stronger with a rewrite, perhaps focusing on some subset of the problem, and much more intuition given to the reader.

---

### Official Review · Reviewer_aFPg · 2021-11-08

**Correctness:** 3
**Technical Novelty And Significance:** 2
**Empirical Novelty And Significance:** 1
**Recommendation:** 3
**Confidence:** 4

**Main Review:**

I was called to do an urgent review on this submission and I did not have time to check the mathematical details in full.
In general I do not think the paper is well-written, as many notations are directly referred to without definition such as \mathbb{\Lambda}, \mathbb{\Pi}, d_0, d_{d_0}, d^{\pi}, R_{\mathcal{W}}(M).
It is difficult to catch the main idea of this paper, and it is dubious if the proposed algorithm can be implemented in practice. Specifically,
- how should one compute $\nabla F_{G,\beta}$ in practice?
- How does one know that a function $f$ which satisfies the definition 2 (function aggregation) exists? How does one know that a policy $\lambda$ that satisfies the definition 4 (shrub) exists?
- In practice, is there a way to estimate policy completeness as defined in Definition 9?
- In definition 10 (distribution mismatch), is there any assumption made to ensure that $C_\infty \ne \infty$?
- Any explicit comparison between the proposed boosting-enhanced learning algorithms and no-boosting RL learning algorithms?

The meaning of notations is inconsistent from place to place.
- $P$ in theorem 11?

**Summary Of The Paper:**

This paper attempts to deploy the boosting technique in the supervised learning on policy learning in the RL setting. The author suggests a protocol to do policy aggregation from weak policies through the quantity $\textit{the extension operator}$, and presents the probabilistic convergence guarantee for the learning process.

**Summary Of The Review:**

I suggest the paper be added more clarification of notations and interpretation of main results. The paper is certainly not ready to be published due to its lack of clarity and unrefined narrative.

---

> ### Author Response · Authors · 2021-11-19
> **Response**
>
> We thank the reviewer for reading the manuscript and for their feedback and suggestions. We will clarify the comments pointed out by the reviewer in the final version.
>
> - Regarding notation, most of the terms that were mentioned by the reviewer are in fact all defined in the paper:
>   - \mathbb{\Lambda} - “a shrub” see Definition 4.
>   - \mathbb{\Pi} - a “policy tree” see Definition 5.
>   - d_0 - a starting state distribution (see Definition of MDP)
>   - d_{d_0}, d^{\pi} - discounted state-visitation distribution, see pg. 4.
>
>    However, indeed some notations are not thoroughly explained, and we will correct that in the final version. Moreover, we agree with the reviewer that the paper is notation-heavy and will add a notation list in the appendix of the paper.
>
>
>
> - Regarding the practicality concerns raised by the reviewer:
>   - “How does one know that ...” - These are not assumptions but merely definitions. We construct these objects via Alg. 1.
>   - “policy completeness” / “distribution mismatch” - these are indeed assumptions we make in the paper. Utilizing such assumptions as the bounded distribution mismatch coefficient C_infty, is required for achieving meaningful results in this context, which is otherwise notoriously difficult to analyze. This follows previous works which also depend on similar assumptions on C_infty, e.g. Agarwal et al. 2019.
>
> - Regarding comparison with no-boosting RL. We stress that this setting cannot be compared with other no-boosting approaches, as it is an entirely different computation and information model. In the boosting setting, the learner is given access to a Weak learner which is assumed to have some non-trivial success guarantees with respect to the target policy class. We stress that although the weak learnability assumptions in this work cannot be easily verified for complex policy spaces (e.g., deep neural nets (DNNs)), we are unaware of any other theory which can make general boosting guarantees without even stricter assumptions. Note that similar "weak learnability" assumptions of DNNs cannot be verified even for renowned classical boosting methods for supervised learning. Nevertheless these methods are used by practitioners and have found practical success. Thus, we view our work as one of the first steps towards provable boosting techniques in RL, which is otherwise notoriously difficult to analyze.

---

### Official Review · Reviewer_Kp4W · 2021-11-08

**Correctness:** 4
**Technical Novelty And Significance:** 4
**Empirical Novelty And Significance:** Not applicable
**Recommendation:** 6
**Confidence:** 4

**Main Review:**

Strength

1. This paper studies boosting methods in RL, which is an interesting topic on its own and has not been explored much from the theoretical perspective yet.
2. The algorithm uses a variant of the Frank-Wolfe method to overcome the non-convexity of the value function (with respect to the policy space). To do so, the analysis relies on recent investigation of the policy gradient algorithm which identify conditions under which the value function is gradient dominated. The application of  the Frank-Wolfe method in RL seems interesting.
3. The writing of the paper is well-structured and very easy to follow. The analysis looks rigorous.


Weakness:
1. The authors only consider weak learner that optimizes a linear function over policy space, which seems restrictive.
2. The analysis does not look very novel given prior work (Hazan & Singh, 2021).

**Summary Of The Paper:**

In this paper, the authors study boosting in RL, i.e., how to convert weak learners into effective policies. The authors provide an algorithm that improves the accuracy of the weak learners iteratively, and the sample complexity and running time do not explicitly depend on the number of states.

In order to overcome the non-convexity of the value function (with respect to the policy space), the authors use a non-convex variant of the Frank-Wolfe method together with recent advances in gradient boosting.

**Summary Of The Review:**

Weighing the strength and weakness of the paper, I am recommending weak accept.

---

> ### Author Response · Authors · 2021-11-19
> **Response**
>
> We thank the reviewer for reading the manuscript and for their feedback and suggestions. We will clarify the comments pointed out by the reviewer in the final version.
>
> - Regarding the WL assumption - we agree it will indeed be interesting to obtain an *improved* guarantee when using a stronger assumption. However, using a *weaker* WL assumption as is the case in this work, strengthens our result. That is, the same result applies for stronger WL assumptions.
>
> - Regarding novelty, we stress that a critical difference from Hazan and Singh 2021 is that their approach applies to *convex* costs, whereas our approach does not require that assumption. Moreover, our results are based on a novel proof for a variant of the Frank-Wolfe optimization technique, adapted to non-convex and gradient dominated function classes, a commonly studied setting in the context of RL.

---

### Official Review · Reviewer_LN8M · 2021-11-11

**Correctness:** 3
**Technical Novelty And Significance:** 3
**Empirical Novelty And Significance:** 3
**Recommendation:** 5
**Confidence:** 3

**Main Review:**

Strengths:
1. The idea of transferring the most recent online boosting technique to RL is interesting.
2. This paper gives solid theoretical analysis of the proposed algorithms.

Weakness:
1. The novelty looks limited as the techniques are very similar to Hazan and Singh 2021. Moreover, this paper does not discuss the major difficulty of designing boosted RL compared to designing boosted contextual bandits, which has already been proposed in existing literature.
2. The presentation has some problems. If this paper aims to deal with the large state space problem in RL, then it misses some important related work in state compression techniques.
3. Although this paper is mainly theoretical, some initial experiments will be beneficial to validate the effectiveness of the proposed algorithms. Hazan and Singh 2021 also gives some experiments.

**Summary Of The Paper:**

This paper studies boosting-like RL algorithms using the most boosting techniques from online learning. It advocates the advantage of the proposed algorithm as the sample complexity's independence of the number of states.

**Summary Of The Review:**

This paper extends the recent online boosting technique to the RL setting. The current presentation has some problems and the technical novelty looks a little bit incremental.

I recommend the authors to (1) discuss the difference between designing boosted contextual bandits and boosted RL;
(2) design some initial experiments to justify the effectiveness of the proposed algorithms.
(3) consider to present the logic of this paper in another way (not motivated by the large state problem in RL).

---

> ### Author Response · Authors · 2021-11-19
> **Response**
>
> We thank the reviewer for reading the manuscript and for their feedback and suggestions. We will clarify the comments pointed out by the reviewer in the final version, as well as missing citations.
>
> - Regarding novelty, we stress that a critical difference from Hazan and Singh 2021 is that their approach applies to *convex* costs, whereas our approach does not require that assumption. Moreover, our results are based on a novel proof for a variant of the Frank-Wolfe optimization technique, adapted to non-convex and gradient dominated function classes, a commonly studied setting in the context of RL.
>
> - Regarding experiments, since the focus of this work is theoretical investigation of boosting algorithms for RL, empirical results were beyond the scope of this work. We agree that it would indeed be very interesting to conduct such experiments in future work. However, we feel that proposing a boosting approach to full RL with provable guarantees is sufficient in terms of innovation for this paper.

---

### Decision · Program_Chairs · 2022-01-20

**Decision:**

Reject

**Comment:**

The paper proposes a boosting algorithm for RL based on online boosting. The main advantage of the result is that the sample complexity does not explicitly depend on number of states. Post rebuttal, some of the reviewers have changed their opinion on the paper. However, overall the reviewers still seem to be on the fence about this paper. Seems like the paper combines the techniques from Hazan Singh’21 along with a frank-wolfe algorithm to deal with non-convex sets but the reviewers seem to view this as not as significant a new contribution.

I see the paper as being interesting but do agree with some of the comments of the reviewers and am leaning to a reject.